# ALL LIFE IS PROBLEM CREATION: LEARNING TO GENERATE ENVIRONMENTS THAT MAXIMIZE PERFORMANCE GAIN

## ABSTRACT

Intelligent agents can achieve mastery not just by learning on well-defined problems, but also by creating their own experiences that maximise learning. While current methods for automatic curriculum generation often rely on heuristics such as task novelty or difficulty, these proxies are often misaligned with the ultimate task. An agent can be endlessly captivated by novel-but-unlearnable environments or stymied by difficult-but-irrelevant challenges. We propose a framework where a generative 'Proposer' agent learns to create environments that explicitly maximise 'Solver' agents' performance gain on a target task. To make the curriculum adaptive, the Proposer is conditioned on the Solver's policy, obtained by probing its behaviour on a small set of diagnostic environments. This conditioning mechanism enables the Proposer to generate a sequence of training environments, targeting the Solver's evolving weaknesses. We validate our approach in maze environments, where our method learns to generate a curriculum of environments that are distinct from the target task distribution. Our experiments demonstrate that this approach accelerates the Solver's learning on both in-distribution and out-of-distribution tasks compared to training directly on the target distribution.

## 1 INTRODUCTION

Much of the history of artificial intelligence has focused on building supervised learning, generative modelling, and planning algorithms to solve well-defined problems. Though this aligns with the philosophical view that "All life is problem solving" Popper (1994), it overlooks the higher-order skill of posing the very problems that are most valuable to solve for building general competence. Indeed, intelligent agents can achieve mastery by solving self-prescribed problems Schmidhuber (2009). For example, a football player may practise brief, targeted drills that bear little resemblance to a full match yet maximise performance in actual games. In this paper, we study machines that learn to propose problems themselves, envisioning agents that improve by designing their own curricula. These agents set and accomplish tasks to maximise their competence in an open-ended way, continually searching for novel and learnable challenges Schmidhuber (2012). This capability would allow an agent to discover questions that lie beyond the boundaries of existing data and learn in a truly unsupervised open-ended fashion.

Recent progress in large language models, while impressive, has already approached the limits of human-generated data in various domains such as software development and reasoning Silver & Sutton (2025). Consequently, state-of-the-art methods are moving beyond supervised finetuning on static human-labelled data to training agents with reinforcement learning in synthetic domains such as math and coding Guo et al. (2025). Furthermore, even when environments are abundant, learning from uncurated experience is often sample inefficient. An expert chess-playing agent, for instance, gains little from repeatedly playing full games against itself Silver et al. (2016) or practicing standard openings it has already mastered. Instead, an ideal agent should learn to imagine specific, challenging endgame puzzles that target its current weaknesses.

This raises a central question: how can an agent learn to generate useful learning experiences for itself? Prior work has proposed several proxies for usefulness, most prominently task novelty Schmidhuber

Figure 1: Overview of our method. We train a generative **Proposer** ($\pi_p$) to create an adaptive curriculum that maximizes the performance of a **Solver** ($\pi_s$). The process is a loop: **(1)** The Proposer is conditioned on the Solver's current policy by probing its behavior on a set of diagnostic tasks. **(2)** The Proposer then generates a batch of $P$ environments at each step $l \in \{1, ..., L\}$ for the Solver to train on. **(3)** Finally, the Proposer receives a reward calculated from the Solver's total performance gain after $L$ steps on a random held-out set of target tasks.

(1991b); Storck et al. (1995); Bellemare et al. (2016); Tang et al. (2017); Pathak et al. (2017), difficulty Sukhbaatar et al. (2017); Zhao et al. (2025b), and compressibility Schmidhuber (2010). While such heuristics can drive exploration, they are limited. Pure novelty-seeking, for example, often rewards unpredictability without regard for learnability, leading to the situation in which agents are endlessly distracted by stochastic but uninformative signals Schmidhuber (2010; 2012). Similarly, maximising difficulty is unreliable: a task may be hard but irrelevant, or so unsolvable that it provides no signal for improvement. Crucially, the utility of any task depends on the student: useful tasks for a novice may be trivial for an expert.

In this work, we study these proxy objectives and propose a more direct approach. We frame problem generation as a principled optimisation problem, where a generative model is explicitly conditioned on the current policy of the learning agent. This conditioning is achieved by probing the agent's behaviour on a small set of diagnostic tasks, allowing the generative model to create a compact representation of the agent's current weaknesses. Its sole objective is to generate environments and tasks that maximise the agent's performance gain on a target distribution of tasks. Importantly, the proposed tasks are not subtasks of the target task and may even be different from the target task distribution, much like a football player practicing toe-bounce drills rather than playing a full 90-minute match.

We demonstrate that agents trained with our method learn faster in both in-distribution and out-of-distribution target environments compared to baselines trained on random target environments. Furthermore, we analyse the emergent curriculum, showing that the generative model learns to create a structured and interpretable sequence of tasks. We study the properties of this curriculum, showing that our agent can learn to recognise what constitutes a useful experience and how it correlates with measures of progress.

## 2 RELATED WORK

Our approach is most related to *curriculum learning*, which selects tasks to guide training, *self-training*, which generates problems that the agent can learn from, and *hierarchical reinforcement learning*, which decomposes complex problems into subgoals and skills.

**Curriculum Learning.** Curriculum learning aims to accelerate training by structuring the order of experiences (Bengio et al., 2009; Narvekar et al., 2020; Portelas et al., 2020). Early works relied on manually designed task sequences. Later, seminal work by Graves et al. (2017) and Matiisen et al. (2019) introduced the teacher–student framework, where a teacher adaptively selects tasks based on the student's progress. A significant body of work (Fan et al., 2018; Katharopoulos & Fleuret, 2018) focusses on optimal data selection, where the teacher learns to filter, reweight, or select the most impactful examples from a fixed dataset. In contrast, our method aims to generate the optimal experience, potentially substantially different from the original training distribution. Based on this idea, the follow-up methods explored various task generation strategies, including *Reverse Curriculum Generation* (Florensa et al., 2017), *GoalGAN* (Florensa et al., 2018), and *ALP-GMM* (Portelas et al., 2019), which adapt task difficulty within a predefined family of environments.

Another important dimension is how the teacher itself is optimised. Most existing approaches rely on short-term or local criteria, such as maintaining a target success rate (Florensa et al., 2018), maximising local learning progress (Portelas et al., 2019), or adversarially generating environments based on immediate regret signals (Dennis et al., 2020; Jiang et al., 2021). While effective, these methods emphasise short-term progress rather than long-term student performance. In contrast, our method remains goal-directed: the teacher adaptively proposes auxiliary tasks —-potentially distinct from the original goal —- but is optimised solely for the long-term success of the student on one predefined complex objective.

**Self-Training.** Another line of related research concerns agents that create their own learning signals, moving beyond fixed datasets or pre-defined reward functions. This concept has roots in the study of intrinsic motivation, where agents are rewarded for exploring novel states or improving their own world models, such as artificial curiosity (Schmidhuber, 1991b). This paradigm includes asymmetric self-play, where a "teacher" agent learns to propose challenging yet solvable goals for a "student" (Sukhbaatar et al., 2018), and frameworks like PowerPlay, which explicitly search for novel and learnable problems to drive open-ended skill acquisition (Schmidhuber, 2012).

More recently, this paradigm has been revitalised in the context of large language models. For example, *R-Zero* and *Absolute Zero* (Huang et al., 2025; Zhao et al., 2025a) show that pretrained large language models can be finetuned with reinforcement learning by autonomously generating, solving, and verifying their own tasks, similarly to self-generated challenges and world models (Schmidhuber, 1992; 2015). Other approaches generate diverse auxiliary tasks, such as asymmetric self-play (Sukhbaatar et al., 2018), where one agent generates challenges for another, and POET (Wang et al., 2019), which co-evolves environments and agents to discover diverse auxiliary problems.

In a related vein, Self-Rewarding Language Models use the model's own judgment to provide reward signals for iterative fine-tuning (Yuan et al., 2024). While our framework shares the spirit of self-training, its objective is fundamentally different. Unlike curiosity-driven or open-ended systems that reward novelty or solvability on a set of tasks, our Proposer learns to condition on the current Solver's abilities and is rewarded exclusively for the Solver's performance gain on an external target objective.

**Hierarchical Reinforcement Learning (HRL).** HRL tackles long-horizon problems by decomposing them into manageable subtasks. Early concepts in HRL involved using recurrent neural networks as subgoal generators, which learned to propose intermediate steps for a reinforcement learning agent (Schmidhuber, 1991a). The *options framework* (Sutton et al., 1999) formalised temporally extended actions, later extended by the *Option-Critic architecture* (Bacon et al., 2017) to enable end-to-end option learning. Other approaches include FeUdal Networks (Vezhnevets et al., 2017), which employ manager–worker structures with abstract subgoals, and unsupervised skill discovery methods such as VIC (Gregor et al., 2016) and DIAYN (Eysenbach et al., 2019), which learn diverse reusable skills for downstream tasks.

Another closely related line of work is goal-conditioned reinforcement learning (GCRL) with methods like Hindsight Experience Replay (Nair et al., 2018; Andrychowicz et al., 2017; Pong et al., 2019; Schaul et al., 2015), which learn from failure by retroactively treating achieved states as successfully completed goals. Unlike GCRL and HRL, which decomposes tasks into subgoals or skills directly tied to the target objective, our teacher proposes auxiliary tasks that may be semantically distinct yet still beneficial for learning (e.g. "kick football" instead of "play 90-minute football game "), thus broadening the training signal while remaining explicitly optimised for the final objective.

## 3 METHOD

Our objective is to train a generative model, the **Problem Proposer** $\pi_p$, which samples environments to maximise the performance of a learning agent, the **Solver** $\pi_s$, on a target task distribution.

We formalise this as a reinforcement learning (RL) problem where the Proposer, $\pi_p$, is an RL agent whose action is to generate a batch of training environments. It receives a reward based on the subsequent performance gain of the Solver, $\pi_s$, on the target task. This section details our framework: we first describe the generative model of the Problem Proposer (§3.1). Since the optimal training curriculum is not static but should adapt to the Solver's evolving capabilities, in (§3.2), we introduce the mechanism for conditioning Proposer $\pi_p$ on the Solver's policy $\pi_s$. We then define

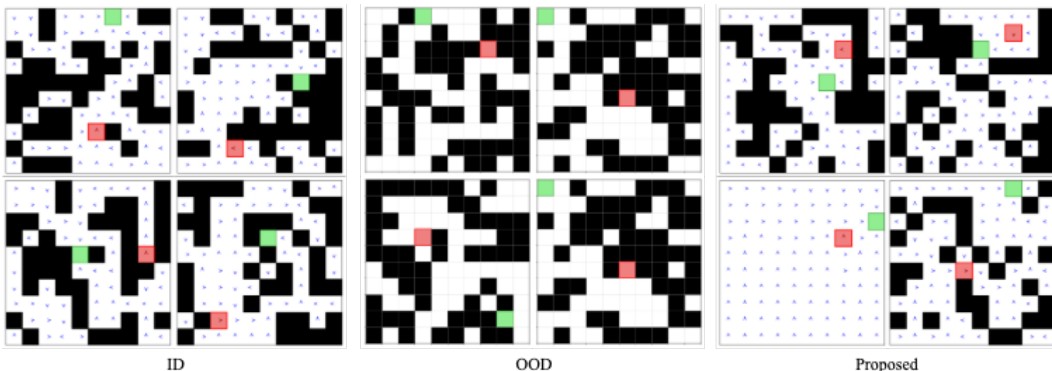

Figure 2: A visual comparison of randomly sampled environments from the **in-distribution (ID)**, **out-of-distribution (OOD)**, and our **Proposer-generated** distributions. The ID and OOD tasks are structurally complex with fixed shortest path lengths of 12 and 14, respectively. In contrast, the Proposer's curriculum (shown here from early in training) consists of visibly simpler environments with significantly shorter path lengths and fewer obstacles. This highlights the Proposer's strategy of generating a distinct distribution of tasks to bootstrap the learning process.

the performance gain reward used to train the Proposer (§3.3) and outline the sequential generation process that allows the curriculum to adapt dynamically (§3.4).

### 3.1 GENERATIVE MODEL

The Proposer network, $\pi_p$, is a generative model that outputs a batch of $P$ problems at each step. The model's objective is to generate a problem distribution that is more effective for training the current Solver than randomly sampling directly from the target distribution. While this approach is similar to training agents in generated worlds Ha & Schmidhuber (2018), a key difference is that our generative model is explicitly optimised via reinforcement learning to be useful for the Solver.

The Proposer's goal is to generate an entire batch of $P$ problems. For example, in the maze navigation task, the Proposer would generate $P$ number of new mazes for the Solver to train and learn from on. Capturing the joint distribution over these environments is desirable as it allows the model to control batch-level properties like task diversity. However, modelling the joint distribution naively (e.g. autoregressively) is computationally expensive for large $P$ (we use $P = 128$).

To maintain efficiency, we model the problems as conditionally independent of each other given the Solver's policy $\pi_s$. The Proposer architecture consists of a decoder that receives two inputs: (1) a conditioning vector representing the Solver's policy $\pi_s$ (detailed in §3.2), and (2) a Fourier encoding of the problem index $p \in \{1, ..., P\}$. The decoder then outputs the categorical parameters for the $p$-th problem. This design allows the Proposer to learn to sample a diverse set of problems within a single batch, as observed in our experiments.

### 3.2 CONDITIONING ON A PROBED SOLVER'S POLICY

The optimal training experience for an agent depends on its current weaknesses. For example, a chess agent that has mastered openings benefits more from practicing complex endgames. To this end, our Proposer $\pi_p$ is conditioned on the current state of the Solver's policy $\pi_s$.

To obtain a representation of the Solver's policy, $\pi_s$, we first execute it on a set of $C$ probe environments sampled from the target distribution. For each environment, we collect both the states (e.g. the 2D mazes) and the corresponding action logits (e.g. probabilities for going up, down, left and right at each cell) produced by the Solver's policy network. These states and logit tensors are then concatenated and processed by a convolutional neural network Fukushima (1980); LeCun et al. (1998) to form a fixed-size conditioning vector. This vector serves as the representation of the Solver and is an input to the Proposer network. In our experiments, we randomly sample $C = 16$ probing environments.

Notably, this conditioning mechanism does not require $\pi_p$ to accurately evaluate the Solver's policy or predict its value function, a task known to be difficult Faccio et al. (2022). Instead, $\pi_p$ only needs to recognise patterns indicative of the Solver's performance. For instance, a coach may not know the exact probability of winning a football match but can observe that a player is struggling with a specific skill (e.g. running) and generate problems to target that weakness. Similarly, $\pi_p$ learns to identify where $\pi_s$ performs poorly and proposes relevant problems accordingly.

### 3.3 Training with Performance Gain Reward

We simultaneously train both the Proposer $\pi_p$ and the Solver $\pi_s$ from scratch using Proximal Policy Optimization (PPO) Schulman et al. (2017). The reward for the Proposer is designed to directly optimise for the Solver's performance.

Specifically, we define the Proposer's reward as the **performance gain** of the Solver on a held-out set of target tasks. Let $\pi_s^{(0)}$ be the Solver's policy before a curriculum begins, and let $\pi_s^{(L)}$ be the policy after training for $L$ steps. Let $R_s(\pi, \mathcal{D}_{\text{target}})$ be the expected return of a policy $\pi$ on the target problem distribution $\mathcal{D}_{\text{target}}$. The total gain for the full curriculum is:

$$R_{p,\text{total}} \;=\; R_s\big(\pi_s^{(L)}, \mathcal{D}_{\text{target}}\big) \;-\; R_s\big(\pi_s^{(0)}, \mathcal{D}_{\text{target}}\big) \tag{1}$$

In practice, this expectation is estimated over a fixed validation batch of $E$ problems from $\mathcal{D}_{\text{target}}$. A new validation batch is randomly sampled for each of the Proposer's main policy update steps. However, to reduce the variance of the reward signal, this same batch is used to evaluate the performance gain for all parallel rollouts within that single update step.

To provide a denser learning signal for the Proposer, we distribute this total reward over the $L$-step curriculum. At each step $l \in \{1, ..., L\}$, the Proposer receives an intermediate reward, $r_p^{(l)}$, equal to the marginal performance gain from that single step:

$$r_p^{(l)} \;=\; R_s\big(\pi_s^{(l)}, \mathcal{D}_{\text{target}}\big) \;-\; R_s\big(\pi_s^{(l-1)}, \mathcal{D}_{\text{target}}\big), \quad l = 1, \dots, L \tag{2}$$

The cumulative return for the Proposer's $L$-step episode, with a discount factor of $\gamma = 1$, is equivalent to the total performance gain, $R_{p,\text{total}}$. This formulation provides a step-by-step learning signal that facilitates more stable training without introducing greedy bias. This reward can be negative if a particular step causes the Solver's performance to drop.

Our objective contrasts with methods that reward performance on the generated tasks themselves Zhao et al. (2025a). By focussing on the target distribution, we ensure that the curriculum remains grounded in the ultimate goal. The number of Solver updates, $L$, is a key hyperparameter. A small $L$ may lead to myopic curricula that maximise immediate gain, while a larger $L$ encourages curricula with better long-term benefits at a higher computational cost.

### 3.4 Step-by-step generation

A potential design choice is to have $\pi_p$ generate a full curriculum of $L \times P$ problems at once. However, Solver's learning trajectory is inherently ambiguous and stochastic (e.g. solver's policy and its learning progress are hard predict) and therefore a static approach will be suboptimal. Instead, we employ a sequential generation process where the Proposer adapts to the Solver's progress at each step of the inner training loop. The process is as follows: at each curriculum step $l \in \{1, ..., L\}$, the Proposer $\pi_p$ conditions on the Solver's current policy $\pi_s^{(l)}$ and generates a batch of $P$ problems. The Solver then performs one or more gradient updates on this batch to produce an updated policy, $\pi_s^{(l+1)}$. This updated policy is then used to condition the Proposer for the next step. This iterative loop, summarised in Figure 1, allows the curriculum to be highly responsive to the stochastic Solver's learning trajectory.

## 4 Experiments

Our experiments are designed to investigate four key questions. First, does a proposer-generated curriculum accelerate learning and improve final performance compared to standard RL? Second, is

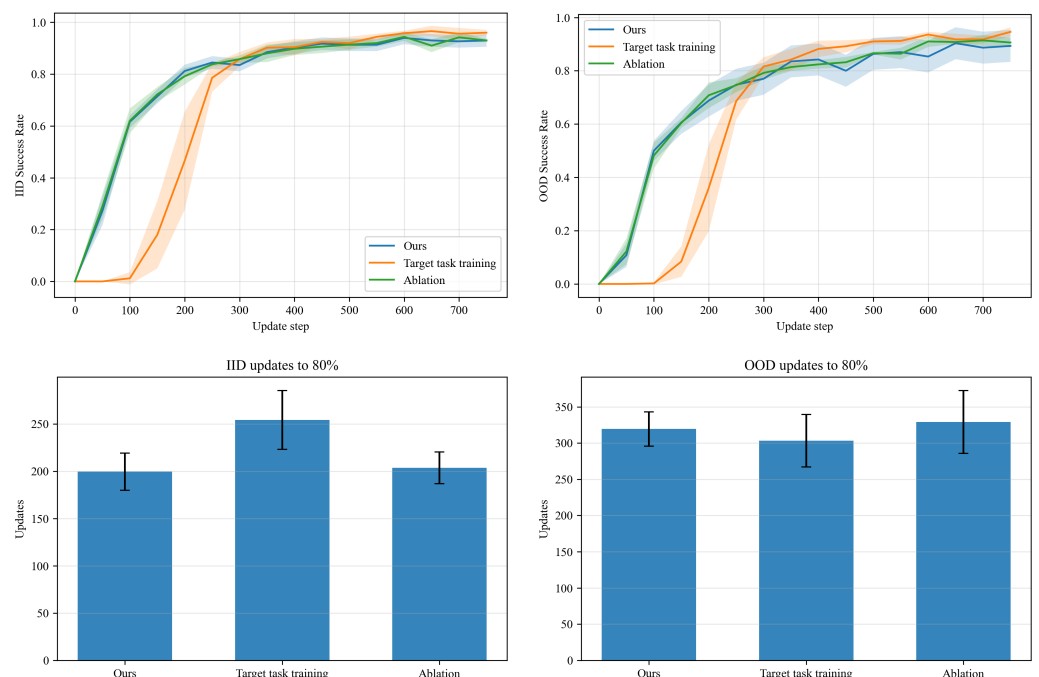

Figure 3: **Quantitative Results.** (Top Row) Learning curves showing Solver success rate vs. training update steps for in-distribution (ID) and out-of-distribution (OOD) evaluation tasks. Our method and the ablation show significantly faster initial learning compared to the target task training baseline. (Bottom Row) Sample efficiency, measured as the number of updates to reach 80% success rate. Our method is notably more efficient on the ID task.

the performance gain reward essential, or can a simpler heuristic like task difficulty achieve similar results? Third, does the emergent curriculum also accelerate learning on out-of-distribution (OOD) tasks? Finally, what are the qualitative characteristics of the curricula our method discovers?

## 4.1 EXPERIMENTAL SETUP

The experiments are conducted in a procedurally generated 2D **maze environment** of size $10 \times 10$, containing a start location, a goal location, and obstacles. The Solver agent selects from four discrete actions (up, down, left, right) and receives a sparse reward of $+1$ only upon reaching the goal, with an episode terminating if a step limit is exceeded. It is rewarded to complete the task in fewer steps by adding a per-step penalty of $\alpha$ (we use $\alpha = 0.01$).

A central element of our experimental design is the use of two narrow and distinct task distributions, created via different underlying data generation mechanisms, to rigorously test our claims. Our primary benchmark, which we refer to as the **in-distribution (ID) target task**, is a narrow distribution of mazes procedurally generated with a fixed shortest path distance of 12 and 40% obstacle density. This distribution serves as the consistent benchmark for evaluating all agents and is used to train the baselines. This setup allows us to verify that our Proposer learns a curriculum that is not merely samples from the target environment distribution, but is verifiably different (e.g. has different number of obstacles or a different optimal path length). To measure generalization, we also introduce a more challenging **Out-of-Distribution (OOD) Target Task** with a path distance of 14 and 50% obstacle density. This distribution allows us to test whether the emergent curriculum enables the Solver to acquire more robust and generalizable skills.

The **Solver** is parameterized by a UNet architecture Ronneberger et al. (2015), which takes the full maze as input and outputs action logits and a value estimate for each cell. For data collection, the policy is executed for a fixed horizon of 25 steps per environment. If an episode terminates early (e.g. by reaching the goal), the agent's position is reset, allowing it to sample multiple trajectories

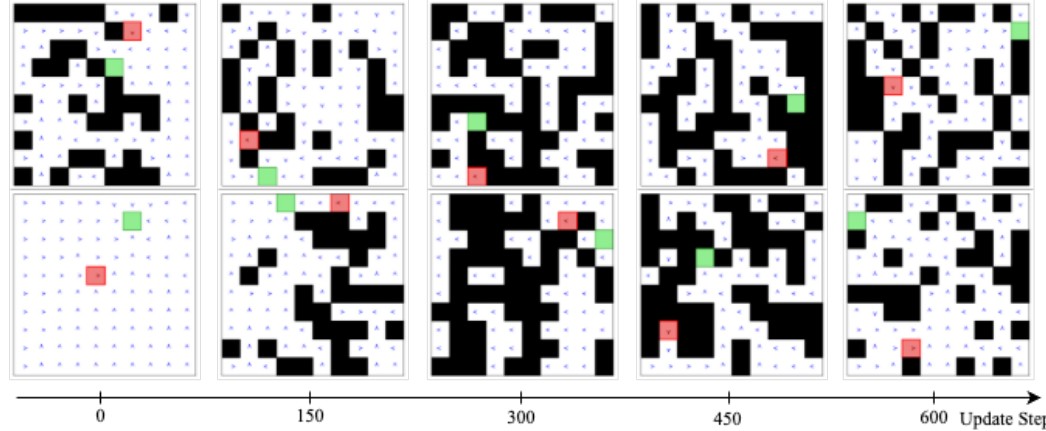

Figure 4: **Visualization of the emergent curriculum generated by our Proposer at different training update steps (columns).** We show the first two environments (rows) from each batch of $P = 128$. The curriculum clearly progresses from simple mazes with short solution paths in early training (left) to more complex and diverse challenges in later stages (right). Note that these generated environments, with their varying path lengths and obstacle counts, are demonstrably different from the fixed target task distribution.

from the same maze within this horizon. The collected experience is then used to perform eight PPO update steps with a learning rate of $2.5 \times 10^{-4}$, a discount factor $\gamma = 0.99$, a generalized advantage estimation (GAE) parameter $\lambda = 0.95$, and a clipping parameter $\epsilon = 0.2$. The loss function includes an entropy coefficient of $0.01$, a value function coefficient of $0.5$, and the gradient norm is clipped at $1.0$. We train all agents for a total of 750 such data collection and update cycles.

The **Proposer** agent is parameterized by a convolutional neural network that receives the Solver's policy representation and outputs the categorical parameters for a batch of $P = 128$ mazes. At the start of each curriculum episode, the Proposer is conditioned on the Solver by probing its policy on a set of 16 probing environments. The Solver then trains for a period of $L = 50$ update steps on the curriculum generated by the Proposer. After these 50 steps, the Proposer's reward is calculated as the Solver's performance gain, estimated over a fixed, held-out set of 75 target tasks. This entire sequence constitutes one data collection rollout for the Proposer. We gather experience from 16 such parallel rollouts before updating the Proposer's policy for 3 updates, with each update consisting of 16 epochs. The remaining PPO hyperparameters, such as the learning rate and discount factor, are identical to those used for the Solver.

We compare our method, **Proposer (Perf. Gain)**, against two primary baselines. The **Random Target Task** baseline represents the standard RL approach, where the Solver trains on problems sampled directly from the ID Target Task distribution. The second, **Proposer (Task Difficulty)**, is a crucial ablation of our method. It employs the identical Proposer-Solver architecture but rewards the Proposer for generating tasks that are maximally difficult for the Solver, rather than those that maximize performance gain.

All methods are evaluated on their **sample efficiency**, defined as the mean number of updates to achieve an 80% success rate, and through **learning curves** that plot the mean success rate over the course of training. For statistical robustness, all results are averaged across 5 runs with different random seeds, and the learning curves are presented with 95% confidence intervals.

## 4.2 QUANTITATIVE RESULTS

Our primary finding is that a curriculum generated to maximise performance gain accelerates learning. As shown in the learning curves in Figure 3 (top row), our method (**Ours**) and the ablation (**Ablation**) begin learning almost immediately. In contrast, the baseline that trains only on the target task (**Target task training**) experiences a long initial phase of stagnation, failing to achieve any meaningful progress for the first 150 update steps. This demonstrates the effectiveness of a generated curriculum

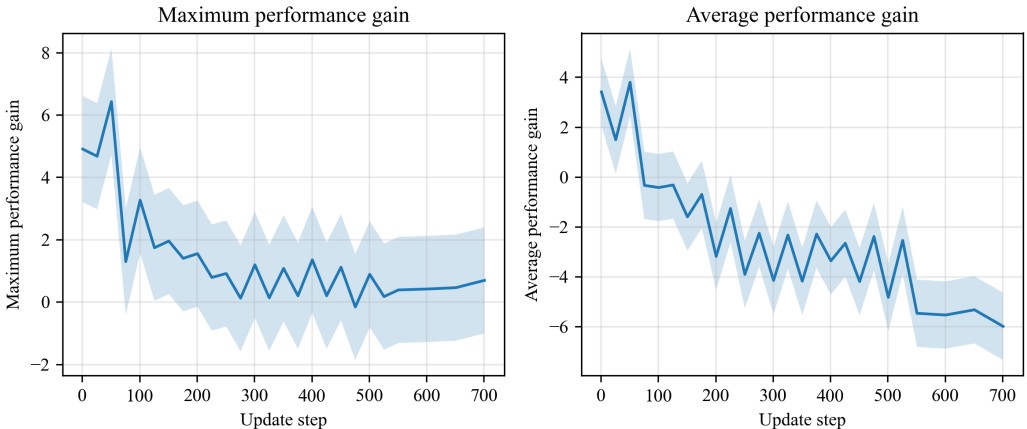

Figure 5: A comparison of the Proposer's effectiveness over time, measured by both the maximum (left) and average (right) performance gain. Early in training, both metrics are high, showing the Proposer consistently generates useful curricula. As training progresses, a clear divergence emerges: the maximum gain remains positive, which demonstrates that the Proposer still successfully finds and generates rare, high-impact environments. In contrast, the average gain diminishes and becomes negative, indicating that useful problems become increasingly difficult to find as the Solver masters the task.

in bootstrapping the learning process, an advantage that holds for both in-distribution (ID) and out-of-distribution (OOD) evaluation.

The sample efficiency results, shown in Figure 3 (bottom row), further quantify this advantage. On the ID task, our method requires substantially fewer training updates to reach the 80% success threshold compared to the target task training baseline. Our method also shows a slight, though not statistically significant, improvement in initial learning speed over the difficulty-based ablation, whose performance lies within the confidence intervals of our own. On the more challenging OOD task, all methods eventually converge to a high success rate and exhibit comparable sample efficiency. The key advantage of our approach, therefore, lies in its ability to dramatically speed up the initial acquisition of skills.

### 4.3 ANALYSIS OF THE EMERGENT CURRICULUM

To understand the mechanism behind these results, we analyse the curricula generated by our Proposer. In Figure 2, we provide a visual comparison between environments sampled from the target distributions and those generated by our Proposer early in training. The figure displays four randomly sampled mazes from the in-distribution (ID) target task, the out-of-distribution (OOD) target task, and our generated curriculum. A clear structural difference is apparent: while the ID and OOD tasks are complex, with fixed shortest path distances of 12 and 14 respectively, the Proposer's environments are visibly simpler. These generated tasks often feature fewer obstacles and significantly shorter path lengths (e.g. 3 to 5), which visually confirms that our method discovers a distinct and simpler distribution of problems to bootstrap the learning process.

Figure 4 provides a qualitative snapshot, visualising mazes generated at different points in the Solver's training. Note that our method samples $P = 128$ environments jointly; here, we choose to visualise the first 2. The curriculum begins with simple, open mazes with short solution paths. As the Solver improves, the Proposer adaptively increases task complexity, introducing more intricate obstacle patterns and longer paths that are demonstrably different from the fixed target task.

We hypothesise that this emergent strategy is effective for two primary reasons. First, the initial phase of simple short-distance tasks likely provides a denser reward signal, which could be crucial for bootstrapping the learning process. This may help the Solver overcome the severe challenge of reward scarcity that causes the baseline agent to stagnate (as seen in Figure 3). Second, a potential

benefit of these shorter initial episodes is that they allow more trajectories to be sampled within a fixed computational budget. This, in turn, might lead to lower variance gradient estimates and more stable policy updates during the critical early stages of training.

This focus on generating useful learning environments is reflected in the Proposer's own learning process, as shown in Figure 5. Initially, the Proposer consistently generates environments that yield high performance gains on average (right plot), effectively bootstrapping the Solver. As training progresses, a divergence between the maximum and average gain emerges. The maximum gain remains positive, demonstrating that the Proposer continues to successfully identify and generate useful environments even for a proficient Solver. In contrast, the average gain diminishes and becomes negative, highlighting that finding useful problems becomes increasingly difficult and that a generic challenging environment is often detrimental to an expert agent. This confirms that the Proposer learns not just to generate problems, but to conduct a targeted search for the specific experiences that are most beneficial at each stage of learning.

## 5 CONCLUSION

In this work, we introduced a new framework for curriculum generation that moves beyond common heuristics, such as novelty or difficulty. Our method trains a generative Proposer agent to create environments by directly optimising for the Solver agent's performance on a target task. A key component of our approach is its adaptive nature, achieved by conditioning the Proposer on a representation of the Solver's current policy. This representation is efficiently obtained by probing the Solver's behaviour on a small set of diagnostic tasks, allowing the curriculum to be tailored to the agent's weaknesses. Our experiments demonstrated that this goal-directed curriculum generation leads to significantly accelerated learning on both in-distribution and out-of-distribution tasks.

**Limitations.** The primary limitation of our method is the computational cost associated with the Proposer's reward calculation. The reward is based on the Solver's performance gain after training for $L$ steps, requiring a full inner-loop training and evaluation cycle to compute a single reward signal for the Proposer. This introduces a trade-off: a small $L$ reduces computational overhead but may lead to myopic, greedier curricula, while a large $L$ provides a more farsighted training signal at a significantly higher cost.

**Future Work.** Looking forward, our work opens several exciting avenues. While our framework is focused on a predefined target task, a long-term research goal is to investigate its asymptotic properties in a more open-ended setting: Can such a system continually expand its capabilities by defining its own sequence of ever-more-ambitious goals? Another promising direction is to move beyond training from scratch and explore how this framework can be used to fine-tune large, pre-trained models, generating targeted data to elicit or enhance specific capabilities.

**Ethics Statement.** As in all reinforcement learning models, our approach has a potential for maximising an objective in an unintended way. For example, our proposed method searching for problems in an open-ended way could potentially lead the model to discover environment settings that cause misalignment or harm to other agents, including humans. However, our method is strictly trained and validated in synthetic maze environments and does not currently have any direct path to such high-risk applications.

**Reproducability Statement.** We are committed to the full reproducibility of our experimental results. The core methods and setup are fully disclosed in the paper and in the supplementary material, including a detailed description of the environment, agent architectures, task distributions, baselines, and all key PPO hyperparameters. Our results are averaged over 5 random seeds, with 95% confidence intervals provided in the figures. As the environment is procedurally generated, we will provide the complete code for both the environment and our proposed method upon acceptance.

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

# A    SUPPLEMENTARY MATERIAL

This supplement provides additional details on the network architectures, training procedures used in our experiments, and additional results.

## A.1    HYPERPARAMETER DETAILS

Table 1 lists the full set of hyperparameters used for our experiments.

Table 1: Hyperparameters for the Solver and Proposer agents.

| Parameter | Value | Description |
|-----------|-------|-------------|
| **PPO Algorithm** | | |
| Learning Rate | $2.5 \times 10^{-4}$ | Adam optimizer learning rate |
| Discount ($\gamma$) | 0.99 | Reward discount factor |
| GAE Lambda ($\lambda$) | 0.95 | Generalized Advantage Estimation lambda |
| Clipping ($\epsilon$) | 0.2 | PPO clip range |
| Entropy Coeff. | 0.01 | Weight of the entropy bonus |
| Value Function Coeff. | 0.5 | Weight of the value loss |
| Max Grad Norm | 1.0 | Gradient clipping threshold |
| PPO Epochs | 8 | Number of PPO epochs step |
| Update Steps | 750 | Update steps in the outer Solver's training loop |
| **Proposer-Specific** | | |
| Lookahead ($L$) | 50 | Solver steps per Proposer reward |
| Probe Environments ($C$) | 16 | Mazes for Solver conditioning |
| Evaluation Environments ($E$) | 75 | Mazes for performance gain eval |
| Parallel Rollouts | 16 | Proposer rollouts per update |

## A.2    SOLVER IMPLEMENTATION

**Architecture.**    The Solver's policy and value functions are jointly parameterized by a U-Net architecture Ronneberger et al. (2015), which processes the entire $H \times W$ maze to produce a dense per-cell policy and value map. The network takes observations of shape $[B, H, W, C]$ as input. The backbone is a U-Net with a depth of 2, using DoubleConv blocks with GroupNorm and ReLU activations. The encoder progresses through channel dimensions of $64 \to 128 \to 256$ with max-pooling for downsampling, while the decoder uses bilinear upsampling and skip connections. The network terminates in two separate 1x1 convolutional heads: a policy head producing per-cell action logits of shape $[B, A, H, W]$ (where $A = 4$ actions), and a value head producing a value map of shape $[B, 1, H, W]$. To obtain the action logits and value for the agent's current state, we index these output tensors at the agent's cell coordinates. All layers are initialized orthogonally.

**Training.**    The Solver is trained using Proximal Policy Optimization (PPO). At each update step, we collect experience by executing the policy in a batch of $P$ environments for a fixed horizon of 25 steps. If an episode terminates early (e.g. the goal is reached), the agent's position is reset, allowing it to sample multiple trajectories from the same maze within this horizon. The collected experience is then used to perform 8 PPO update epochs. Key hyperparameters include a learning rate of $2.5 \times 10^{-4}$, a discount factor $\gamma = 0.99$, a GAE parameter $\lambda = 0.95$, and a PPO clipping parameter of $\epsilon = 0.2$.

## A.3    PROPOSER IMPLEMENTATION

**Architecture.**    The Proposer is a convolutional neural network Fukushima (1980); LeCun et al. (1998) that outputs categorical actions corresponding to discretized environment parameters. Each action specifies a bin for the obstacle fraction ($M$ bins) and a bin for the shortest-path distance ($N$ bins), resulting in an action space of size $M \times N$.

The conditioning process, which provides the Solver's state to the Proposer, is as follows. First, we probe the Solver's policy $\pi_s$ on a batch of $C$ mazes sampled from the target distribution. For

each maze, we obtain the full policy logit map $[A, H, W]$ and concatenate it with the maze state channels. This combined tensor is processed by a CNN encoder. The resulting feature maps from all $C$ examples are then aggregated via average pooling to produce a single context vector $g \in \mathbb{R}^{128}$. To generate a diverse batch of $P$ problems, we append a 16-band Fourier encoding Vaswani et al. (2017) of each problem index $p \in \{1, \ldots, P\}$ to the context vector $g$. This combined vector is then passed through a shared 2-layer MLP to produce the action logits for each of the $P$ problems.

**Performance-Gain Estimator.** In practice, the stepwise performance gain is estimated on a fixed validation batch $\mathcal{E}$ of $E$ problems: $r_p^{(l)} = \frac{1}{E} \sum_{e \in \mathcal{E}} \left[ r(\pi_s^{(l)}, e) - r(\pi_s^{(l-1)}, e) \right]$. The sum of these stepwise gains, $\sum_{l=1}^{L} r_p^{(l)}$, is an unbiased estimator of the total true gain, $R_{p,\text{total}}$. Using a fixed batch $\mathcal{E}$ for all $L$ steps within a single Proposer update is a key variance reduction technique. To prevent the Proposer from overfitting to this specific batch, we resample a new batch $\mathcal{E}$ for each main Proposer update.

**Training.** The Proposer is also trained with PPO. An episode for the Proposer consists of the Solver training for $L = 50$ steps. At each step $l \in \{1, \ldots, L\}$, the Proposer receives a reward equal to the Solver's marginal performance gain on a held-out set of target tasks. The total return for the episode is the undiscounted sum of these stepwise gains. We collect experience from 16 such parallel Proposer rollouts before performing 3 policy updates, with each update consisting of 16 epochs. The PPO hyperparameters (learning rate, $\gamma$, etc.) are identical to those used for the Solver.

## A.4 BASELINES IMPLEMENTATION

To ensure a fair comparison, the Solver agent in all baselines uses an identical architecture and set of training hyperparameters to our main method. The **Target task training** baseline trains this Solver directly on environments sampled from the ID Target Task distribution. Our ablation, **Proposer (Task Difficulty)**, uses the same Proposer-Solver architecture as our proposed method, with the sole modification being a reward function based on task difficulty instead of performance gain, where the Proposer's reward is calculated as one minus the Solver's success rate on the generated batch of environments. This incentivizes the Proposer to generate tasks that the current Solver finds maximally difficult.

## A.5 COMPUTATIONAL RESOURCES

Our experiments were conducted on a heterogeneous cluster of consumer-grade GPUs, including NVIDIA V100, 3080, and A5000 models. A single training run for our method completes in approximately 10 hours on 8 GPUs. The Proposer's rollouts, while computationally intensive, are independent and highly parallelizable.

