# OpenReview forum: "All Life is Problem Creation: Learning to Generate Environments that Maximize Performance Gain"
_ICLR.cc/2026/Conference — ICLR 2026 Conference Withdrawn Submission_

### Official Review · Reviewer_h4DL · 2025-10-16

**Soundness:** 1
**Presentation:** 3
**Contribution:** 1
**Rating:** 4
**Confidence:** 4

**Summary:**

This paper proposes a framework for automatic curriculum generation in reinforcement learning. Instead of relying on common heuristics like task novelty or difficulty, the authors train a generative "Proposer" agent to create environments that **directly maximize the performance gain** of a "Solver" agent on a specified target task distribution.

The Proposer is conditioned on the Solver's current policy, which is represented by probing its behavior on a small set of diagnostic environments. This allows the Proposer to dynamically generate curricula that target the Solver's evolving weaknesses.

The authors validate their approach in procedurally generated maze environments, demonstrating that their method accelerates learning and improves sample efficiency on both in-distribution and out-of-distribution tasks compared to training directly on the target distribution or using a difficulty-based curriculum.

**Strengths:**

- **Originality**: The primary strength is the novel and principled formulation of the curriculum generation objective. By directly optimizing for the Solver's performance gain on a target task, the method avoids the pitfalls of potentially misaligned heuristics like novelty or difficulty. This is a significant conceptual advance for the field.

- **Methodological Quality**: The Proposer-Solver framework is well-designed. The adaptive nature of the curriculum, achieved by conditioning the Proposer on a representation of the Solver's policy, is a key strength that allows the system to target the Solver's specific weaknesses as they evolve.

- **Clarity**: The paper is written with good clarity.

**Weaknesses:**

- **Computational Cost**: The most significant weakness, which the authors acknowledge, is the computational expense. Calculating the Proposer's reward requires running an entire inner loop of Solver training for $L$ steps and then evaluating it. This "inner-loop" optimization structure can be prohibitively expensive and may limit the practical application of this method to more complex tasks or larger models where training steps are already costly.

- **Limited Experiments**: The validation is performed exclusively in a 2D maze navigation task. While this domain is sufficient for a proof-of-concept and allows for clear visualization, the simplicity of the environment raises questions about the method's scalability and generalizability to more complex, high-dimensional, or continuous control tasks.

- **Sensitivity to Hyperparameter L**: The lookahead window $L$ is a critical hyperparameter that balances computational cost and the farsightedness of the curriculum. The paper mentions this trade-off but does not include a sensitivity analysis or ablation study on $L$.

**Questions:**

- The computational cost of the inner training loop for reward calculation is a major hurdle. Have you considered any methods to make the process more efficient? For instance, could a separate model be trained to predict the Solver's performance gain given the current policy representation and a proposed batch of environments, thus acting as a surrogate reward function?

- Could you provide more insight into the choice of $L=50$? How sensitive is the performance of the method to this hyperparameter? Is there a risk that for small $L$, the method devolves into a greedier strategy that is not significantly better than the task-difficulty ablation?

- The probing mechanism uses the full action logit maps from $C=16$ environments to represent the Solver's policy. How was this number of probe environments determined? Have you experimented with more compact representations of the Solver's policy (e.g., using only performance metrics like success rate/return on the probe set, or a flattened vector of the network weights) as the conditioning for the Proposer?

---

### Official Review · Reviewer_QAzU · 2025-11-01

**Soundness:** 2
**Presentation:** 2
**Contribution:** 1
**Rating:** 2
**Confidence:** 4

**Summary:**

This paper proposes a framework where a generative Proposer agent learns to create training environments that explicitly maximize the Solver’s performance gain on a target task. Unlike prior curriculum generation methods based on heuristics such as novelty or difficulty, the Proposer is trained via reinforcement learning to generate adaptive curricula conditioned on the Solver’s policy, obtained by probing its behavior on diagnostic environments. Experiments in a procedurally generated maze domain demonstrate improved sample efficiency and faster convergence compared to training directly on target environments or with difficulty-based curricula.

**Strengths:**

- The paper clearly articulates the goal of environment generation for performance maximization.
- Conditioning the Proposer on the Solver’s policy via diagnostic probing is a neat idea.
- Results show measurable acceleration in learning and improved sample efficiency.

**Weaknesses:**

- All results are in simple 2D grid mazes, which are not that general or open-ended.
- Weak OOD evaluation: “OOD” tasks are only out of distribution in one aspect of the number of obstacles.
- Limited baseline comparisons.
- The resolution of figures is very low. One tip is that if you make figures in PDF, it will be high-resolution even if you zoom in.

**Questions:**

- How does the trained agent perform on different tasks that might be of a different nature?
- It will be interesting to see how the proposer transfer outside of the task distribution.
- How might the solver generalize to non-grid worlds or wolds with infinite state space?

---

### Official Review · Reviewer_ZZvU · 2025-11-01

**Soundness:** 2
**Presentation:** 3
**Contribution:** 2
**Rating:** 4
**Confidence:** 3

**Summary:**

The paper introduces a framework where a generative proposer learns to create environments that improve a solver agent’s performance on a target distribution of tasks. The proposer is conditioned on the solver’s policy obtained by probing it on a small number of diagnostic tasks and is trained using reinforcement learning (PPO) with a reward equal to the solver’s measured performance gain on held-out target tasks after a fixed number of training steps. The framework is evaluated in a 2D maze environment with fixed maze sizes and obstacle densities. The experiments compare three conditions: training the solver directly on the target distribution,  a proposer ablation where the proposer is rewarded for generating difficult tasks and the proposed Proposer method. The proposed method achieves faster initial learning on the in-distribution maze tasks than the baselines. On out-of-distribution mazes, all methods reach similar final performance. Qualitatively, the generated curricula begin with simpler mazes and later progress to more complex structures.

**Strengths:**

- The objective directly matches the intended outcome, avoiding intermediate heuristics.
- The conditioning procedure is explicitly described and implemented using diagnostic probe environments.
- Empirical evaluation is reproducible: PPO settings, architectures, and evaluation metrics are fully specified.
- Figures and quantitative results demonstrate faster early learning for the proposed method compared with direct target-distribution training.
- The paper explicitly discusses limitations and ethical scope, acknowledging computational overhead and restricted experimental domain.

**Weaknesses:**

- Experimental scope is narrow: only discrete 2D mazes are used, and no additional environments or modalities are tested.
- OOD performance does not exceed baselines, the text notes that all methods converge similarly. Therefore, the claim of improved generalization is not empirically supported.
- Statistical robustness is limited to 5 random seeds, and the text acknowledges that the advantage over the difficulty-based ablation lies within confidence intervals.
- Baseline coverage is minimal: only one ablation (difficulty-based reward) is tested. No comparisons to other adaptive or co-evolutionary curricula are provided.
- Computational demand is high, requiring repeated inner-loop training of the Solver for each Proposer update.
- Assumptions such as conditional independence among generated problems are not evaluated experimentally.
- Reward variance and stability are not quantified, negative gain rewards are mentioned as possible but not analyzed.

**Questions:**

- How sensitive are the reported results to the hyperparameters (probe environments, solver update steps, evaluation batch size)?
- What is the frequency and magnitude of negative stepwise performance-gain rewards during proposer training?
- Did you test alternative formulations that reduce computational cost, such as approximating performance gain with surrogate metrics?
- How does the performance compare when accounting for total wall-clock training time, given the inner and outer PPO loops?
- Are there any observed differences when varying maze sizes or obstacle densities beyond those listed?

---

### Official Review · Reviewer_DCUD · 2025-11-04

**Soundness:** 1
**Presentation:** 2
**Contribution:** 1
**Rating:** 2
**Confidence:** 4

**Summary:**

The paper proposes a method for generating a training curriculum for an RL agent which consists of a proposer (teacher) and a solver (student) policy. The teacher proposes new environments for the student based on the students learning progress.
Also, I'd like to add that citing Schmidhuber monographies 4 times in the introduction and 7 times total gives away which lab this submission is from. In the future, I'd highly recommend to turn this down to remain anonymous.

**Strengths:**

- **S.1:** The writing is mostly clear and accessible.

**Weaknesses:**

- **W.1:** Ablation/impact. I think the fact that the ablation which always picks the most difficult environment performs identically with the proposed method is entirely undercutting this submission.
- **W.2:** Grand claims - unsubstantiated. The paper claims that the method (a) generates a structured and interpretable sequence of environments and (b) importantly, the tasks are not subtasks of the target tasks. For (b): Absolutely not. You are only showing the performance on the 2D maze task. The path being shorter or longer and there being more or fewer obstacles doesn't make what your algorithm generates not a subtask. If your algorithm is able to add a different set of obstacles (e.g. lava tiles that kill the player on contact and give negative reward), that would qualify as "not a subtask" but just changing the trajectory length and obstacle density is very much subtask territory. For (a): again, in the maze task this might be interpretable, but you don't show that this works in any other environment, so you can't make that claim in generality. And even in the maze task, you don't show this convincingly - what I would want to see is a plot showing the path length and obstacle density go up over time and not just 5x 2 random env samples.
- **W.3:** Novelty. With (b) from my previous point unaddressed, there are other methods out there that do something similar, like SS-ADR (https://arxiv.org/abs/2002.07911) or ABC (https://arxiv.org/abs/2101.04882), which also provide more rigorous evaluation across different environments.
- **W.4:** Baselines. There are so many curriculum generation strategies (including the aforementioned ones). Why does this work not compare to any of them? For example, how about the simple auto-curriculum heuristic that's used in the Legged Gym work (https://arxiv.org/abs/2109.11978)?
- **W.5:** Method and implementation details all tangled up. The method section contains many implementation details, like the fact that P=128. In general, I don't mind this - this helps me understand what order of magnitude these HPs are. But then you need to justify in the experiments section why you picked these values and how sensitive they are. If I set P=10, will your method still work? What about P=1000? Same question for C=16 and L=50. Where do these come from and why? If I run this on another environment, how do I pick these?

**Questions:**

- **Q.1:** Not a question but a suggestion. Your algorithm has an inner loop and an outer loop. I think this would be well-illustrated in a pseudocode algorithm block, rather than the hard-to-parse Fig.1
- **Q.2:** If I understand correctly, the embedding of the current student policy is a handful of single steps, right? How do you deal with a high-frequency environment like a robotic one where a single moment in time isn't indicative but you'd need a rollout over multiple steps to tell anything about policy quality?

---

### Note · Authors · 2025-11-12

I have read and agree with the venue's withdrawal policy on behalf of myself and my co-authors.